# Effect of Extracellular Signal-Regulated Protein Kinase 5 Inhibition in Clear Cell Renal Cell Carcinoma

**DOI:** 10.3390/ijms23158448

**Published:** 2022-07-30

**Authors:** Hidenori Kanno, Sei Naito, Yutaro Obara, Hiromi Ito, Osamu Ichiyanagi, Takafumi Narisawa, Tomoyuki Kato, Akira Nagaoka, Norihiko Tsuchiya

**Affiliations:** 1Department of Urology, Yamagata University Faculty of Medicine, 2-2-2 Iida-Nishi, Yamagata 990-9585, Japan; ocean_ape@yahoo.co.jp (H.K.); ito.hiromi@med.id.yamagata-u.ac.jp (H.I.); oichiyan@ab.cyberhome.ne.jp (O.I.); tnari_0623@yahoo.co.jp (T.N.); kato-t@med.id.yamagata-u.ac.jp (T.K.); a-nagaoka@yone-city-hp.jp (A.N.); ntsuchiya@med.id.yamagata-u.ac.jp (N.T.); 2Department of Pharmacology, Yamagata University Faculty of Medicine, 2-2-2 Iida-Nishi, Yamagata 990-9585, Japan; obaray@med.id.yamagata-u.ac.jp

**Keywords:** clear cell renal cell carcinoma, extracellular signal-regulating kinase 5, miR-143

## Abstract

(1) Background: Extracellular signal-regulating kinase 5 (ERK5) has been implicated in many cellular functions, including survival, proliferation, and vascularization. Our objectives were to examine the expression and effect of ERK5 in clear cell renal cell carcinoma (ccRCC). (2) Methods: The expressions of ERK5 and its regulating micro-RNA miR-143 were investigated using immunohistochemistry and quantitative reverse transcriptase PCR in surgical specimens of ccRCC patients. With invitro and in vivo studies, we used pharmacologic ERK5 inhibitor XMD8-92, RNA interference, pre-miR-143 transduction, Western blotting, MTS assay, apoptosis assay, and subcutaneous xenograft model. (3) Results: A strong ERK5 expression in surgical specimen was associated with high-grade (*p* = 0.01), high-recurrence free rate (*p* = 0.02), and high cancer-specific survival (*p* = 0.03). Expression levels of ERK5 and miR-143 expression level were correlated (*p* = 0.049). Pre-miR-143 transduction into ccRCC cell A498 suppressed ERK5 expression. ERK5 inhibition enhanced cyclin-dependent kinase inhibitor p21 expression and decreased anti-apoptotic molecules BCL2, resulting in decreased cell proliferation and survival both in ccRCC and endothelial cells. In the xenograft model, ERK5 inhibitor XMD8-92 suppressed tumor growth. (4) Conclusions: ERK5 is regulated by miR-143, and ERK5 inhibition is a promising target for ccRCC treatment.

## 1. Introduction

An estimated 403,262 new cases of renal cell carcinoma (RCC) were diagnosed worldwide in 2018 (Bray, F. et al. GLOBOCAN2018; https://www.uicc.org/news/new-global-cancer-data-globocan-2018; accessed on 1 September 2020). The most common RCC subtype is clear cell RCC (ccRCC; 70–75%) [1]. Approximately 90% of ccRCCs harbor an inactivation of both von Hippel Lindau (*VHL*) and tumor-suppressor gene alleles [2]. Nephrectomy is used to treat localized RCC; however, half of these patients had metastatic lesions at diagnosis or developed such lesions during the follow-up period [3]. In the case of metastatic RCC (mRCC), systemic pharmacotherapies are used, which target vascular-endothelial growth factors (VEGFs) and mammalian targets of rapamycin complex 1. Most recently, immuno-oncology drugs were developed and established as a treatment for mRCC [4]. These new drugs have been improving the survival of RCC patients; however, mRCC rarely leads to a cure. The identification of new therapeutic targets is warranted to develop better treatment approaches for mRCC.

Extracellular signal-regulated kinase 5 (ERK5), also known as big mitogen-activated protein kinase 1/mitogen-activated protein kinase (MAPK) 7, is a member of the MAPK family [5,6,7]. ERK5 is approximately twice the molecular size of ERK1/2, the best studied MAPK family protein. The kinase domain is encoded by its N-terminal half and shares approximately 50% homology with ERK1/2, while its unique C-terminus encodes two proline-rich regions and a nuclear localization signal. The ERK5 gene knock-out mice are lethal due to cardiovascular defects, which result from the impairment of angiogenesis in ERK5-lacking endothelial cells. ERK5 has been implicated in many cellular functions, including survival, proliferation, and vascularization [8,9]. We previously demonstrated that ERK5 requires catecholamine biosynthesis in neuronal cells [10]. Furthermore, the ERK5 signaling pathway implicates therapy resistance in several human cancers. A recent study revealed that ERK5 is degraded through the ubiquitin–proteasome system in a process mediated by a VHL protein (pVHL) [11] because harboring an inactivation of pVHL, ccRCC should store ERK5. In fact, a previous report demonstrated that 60% of ccRCC patients had overexpression of ERK5 in their surgical specimens [12,13].

ERK5 is reportedly regulated by several miR-143 cancers [14,15,16,17,18,19]. In addition, a previous report demonstrated that RCC patients with low expressions of miR-143 had poor prognoses [20]; however, the correlation between miR-143 and ERK5 in ccRCC remains unknown.

Hence, we investigated ERK5 expressions, the correlation between ERK5 and miR-143, and the functions of ERK5 in ccRCC. In the present study, we demonstrated that (1) ccRCC patients with strong ERK5 expression in their surgical specimens had poor survival; (2) the specimens with decreased miR-143 had higher rates of strong ERK5 expression; (3) ERK5 inhibition suppresses ccRCC-cell and endothelial-cell proliferation and survival in vitro and in vivo.

## 2. Results

### 2.1. ERK5 Expression in Surgical Specimens of ccRCC

First, we analyzed expression of ERK5 in 250 human ccRCC surgical specimens using immunohistochemistry staining. Patient characteristics are shown in Table 1. Two of two-hundred and fifty specimens were not evaluated. Among the remaining 248 specimens, 29 (20%), 139 (55%), and 82 (25%) specimens had negative, weak, and strong expressions of ERK5, respectively. ERK5 expression was not associated with clinical T and M stages (*p* = 0.40 and 0.23, respectively) (Figure 1B,C), whereas a higher rate of high-grade specimens had strong ERK5 expressions (*p* = 0.01) (Figure 1D). The recurrence-free rate and the cancer- specific survival were significantly shorter in the patients with strong ERK5 expression (*p* = 0.020/Figure 1E and *p* = 0.027/Figure 1F, respectively). The multivariate analysis both for recurrence-free rate and cancer-specific survival did not show that ERK5 is an independent prognostic factor (HR [95%CI]; 1.270 [0.616–2.405], 1.099 [0.480–2.514], respectively). To validate that the patients with strong ERK5 expression had shorter survival, we compared overall survival with high and low ERK5 mRNA expression using the TCGA database. As expected, patients with high ERK5 expression level had shorter overall survival (*p* < 0.001).

### 2.2. ERK5 and miR143 in ccRCC Cell Lines

A previous report demonstrated that ERK5 accumulation is caused by an impairment of the pVHL/proteasome system [11]. To validate ERK5 accumulation in ccRCC cell lines, most of which harbor *VHL* inactivation, we examined ERK5 expressions in *VHL* mutant- ccRCC cell lines and a *VHL*-wild type RCC cell line. All examined human RCC cell lines expressed ERK5, especially in A498 and A704 cells. Although a proteasome inhibitor MG132 enhanced ERK5 expression in *VHL* wild-type cell line Caki1, it did not in *VHL*-mutant cell line A498, 786O, 769P, and A704 (Figure 2A).

The expression levels of miR-143 in A498, 786O, Caki1, and A704 were lower than the others, which showed an inverse relationship between the expression levels of miR-143 and ERK5 except 786O (Figure 2B). The induction of pre-miR-143 reduced ERK5 expression in A498 cells (Figure 2C).

### 2.3. miR143 Expression in Surgical Specimens of ccRCC

The range of relative miR-143 expression levels was 0.03–7.66. Of 48 cases that examined miR-143 expression levels, 14 (29.2%) and 34 (70.8%) cases decreased and were normal or had increased miR-143, respectively (Figure 2D). The specimens with decreased miR-143 had higher rates of strong ERK5 expression (*p* = 0.0491) (Figure 2D).

### 2.4. ERK5 Inhibition in ccRCC Cells

To examine the effect of ERK5 inhibition, siRNA for ERK5 was conducted in A498 cells. The transient knockdown of ERK5 increased cell-cycle-dependent kinase inhibitor p21 and decreased anti-apoptotic molecule BCL2 (Figure 3A).

To further confirm the role of ERK5 and investigate the potential for therapeutic target in ccRCC, we used the small inhibitor XMD8-92, which inhibits ERK5 kinase activity (Figure 3B). First, we conducted an MEF2C reporter assay to validate the effect of XMD8-92; MEF2C is a transcription factor that was directly activated by ERK5 [21]. We found that 1.25–10 μM XMD8-92 suppressed MEF2C-dependent transcription in dose–dependent manner (Figure 3C). Western blotting analyses showed that XMD8-92 increased p21 and p27 expression and decreased BCL2, in agreement with the siRNA results (Figure 3D). Moreover, we found that XMD8-92 produced cleaved PARP (Figure 3D) and increased the sub-G1 population from 6.4% in the control to 39.6% in XMD8-92-treated cells (Figure 3E), which indicated an increase in apoptotic cells. To confirm that XMD8-92 induces apoptosis, we investigated the mitochondrial membrane’s potential using TMRE and caspase activity. XMD8-92 reduced TMRE-positive cells and increased caspase 3/7-positive cells (Figure 3F). In addition, we also use another ERK5 inhibitor, XMD17-109, to confirm the effect by pharmacological inhibition. XMD17-109 also showed an increase in apoptotic cells (Figure 3G). Investigating the IC_50_ of XMD8-92 in A498 (*VHL* mutant, low miR-143 expression, and strong ERK5 expression human ccRCC cells, Figure 3H), Caki1 (VHL wild type, low miR-143 expression, and weak ERK5 expression human RCC cells, Figure 3I), 769P (VHL mutant, high miR-143 expression, and weak ERK5 expression human ccRCC cells, Figure 3J), HUVEC (human endothelial cells, Figure 3K), and HRCEpC (normal urothelial cells, Figure 3L), the values were 9.6 μM, 26.5 μM, 28.6 μM, 7.4 μM, and 45.1 μM, respectively. The high ERK5 expression ccRCC cell line A498 and endothelial cell HUVEC had lower IC_50_ than low ERK5 expression RCC cell lines Caki1 and 769P and normal kidney cell line HRCEpC. The IC_50_ of another ERK5 inhibitor XMD17-109 in A498 was 1.3 μM (Figure 3G).

### 2.5. Effect of ERK5 Inhibitor XMD8-92 in Xenograft Model

Using the subcutaneous A498 xenograft model, we examined the effect of XMD8-92 in vivo. We found that the injection of XMD8-92 suppressed tumor growths (*p* = 0.0027) (Figure 4A). There was no weight loss in the mice during treatment (data not shown). Moreover, the injection of XMD8-92 decreased Ki67-positive cells that are markers for cell proliferation (Figure 4B) and CD34, which is a marker for neovascularization (Figure 4C), in a dose-dependent manner.

## 3. Discussion

ERK5 is the most recently discovered MAPK family protein [22]. Its substrates include serine/threonine kinase ribosomal S6 kinase, transcription factors c-Fos, Ankrd1, and MEF2c [21,22,23,24]. ERK5 positively regulates cell proliferation via p21 inhibition and cell survival via BCL2 expression in endothelial cells [25,26,27,28,29]. Several reports showed that strong ERK5 expression indicates worse prognoses in breast, prostate cancer, and ccRCC [12,13,30,31]. In addition, recent studies demonstrated ERK5 as a potential therapeutic target in leukemia, multiple myeloma, breast, liver, prostate, and pancreatic cancer [25,32,33,34,35,36,37,38,39]. Nevertheless, the role of ERK5 in ccRCC remains unknown. We showed that strong ERK5 expression indicates worse prognoses in ccRCC, in agreement with the previous study. In addition, ERK5 expression is associated with the grade but not with the clinical stage (Figure 1B–D). Moreover, our study first demonstrated in ccRCC that (1) ERK5 is regulated not only by pVHL but also by miR-143; (2) ERK5 inhibition suppresses cell viability in highly ERK5-expressed ccRCC cell and endothelial cells; (3) ERK5 inhibition suppressed the expression of anti-apoptotic protein BCL2 and induced apoptosis; (4) ERK5 inhibitor XMD8-92 suppressed tumor growth in xenograft model.

ERK5 is polyubiquitinated by pVHL, which leads to proteasomal degradation [11]. Our study also showed that proteasome inhibitor MG132 did not change the ERK5 expression level, which supports the previous report. In the surgical specimens of this study, however, there were only 25% ccRCC specimens with strong ERK5 expression and as many as 20% without its expression. Nevertheless, beyond 95% of ccRCC, it should harbor *VHL*- or *VHL*-related gene inactivation [2]. These results imply that ERK5 expression in ccRCC should also be regulated by other mechanisms.

A previous report demonstrated that miR-143 suppresses ERK5 in acute myeloid lymphoma, breast cancer, HeLa cell, bladder cancer, and adipose tissue-derived stromal cells [14,15,16,17,18,19]. Other reports demonstrated that miR-143 suppressed cell proliferation via hexokinase-2 and K-RAS and that miR-143 expression level is inversely correlated with early recurrences in RCC [20,40,41]. We demonstrated that the induction of pre-miR-143 suppressed ERK5 expression in ccRCC cells. In addition, low miR-143 expression levels have reverse correlations with ERK5 expression levels in both human ccRCC cell lines and surgical specimens. These results indicate that ERK5 regulation is one of the rationales for tumor suppression by miR-143 in ccRCC. Meanwhile, there should be unknown mechanisms that regulate ERK5 other than pVHL and miR-143 because 786O has low ERK5 expression, despite the presence of the *VHL* mutant and low miR-143 expression levels. Further studies are needed to investigate other mechanisms of ERK5 regulation than pVHL and miR-143.

In the in vivo study, ERK5 inhibitor XMD8-92 suppressed CD34 positive cells, which reflect neovascularization and Ki67, which reflect cell proliferation. A previous study demonstrated that ERK5 gene knock-out mice are lethal due to cardiovascular defects, which results from an impairment of angiogenesis in ERK5-lacking endothelial cells [27]. Another report demonstrated that VEGF activates ERK5 in endothelial cells [42]. Our results also demonstrated that ERK5 inhibition suppressed endothelial cells. These results indicate that XMD8-92 not only directly inhibits cancer cells but also targets cancer angiogenesis. Moreover, ERK5 inhibition might contribute to overcoming anti-VEGF therapy resistance in ccRCC. Although the anti-VEGF therapies are prevalent in metastatic ccRCC treatments, almost all patients develop resistances against them [1]. One of the reasons for the resistance is that other growth factor axes than VEGF, e.g., fibroblast growth factor, hepatocyte growth factor, and hepatocyte growth factor/cMet receptor, are strongly expressed and work for angiogenesis through the anti-VEGF treatment [43,44]. Since ERK5 is a molecule at down-stream of the growth factor, ERK5 inhibition can evade interactions between growth factors. Further studies will be needed to investigate if ERK5 inhibition overcomes the resistance against anti-VEGF therapy.

ERK5 activation requires the stimulation by MEK5. Generally speaking, the signal pathway of MEK5/ERK5 is activated by environmental stress, growth factor, and cytokine stimulation [27,33,42,45,46]; however, the mechanism of ERK5 activation in ccRCC is unclear. Further studies will be needed to investigate the mechanism for ERK5 activation. One of the other limitations is the potential bias of animal study. We did not use the strategy to minimize potential confounders, such as the order of treatment and measurements or animal cage location.

## 4. Materials and Methods

### 4.1. Clinical Specimens and Tissue Collection

A total of 250 ccRCC specimens were collected from patients who had undergone nephrectomy in Yamagata University between 2006 and 2012. Patients with adjuvant targeted therapy and non-ccRCC were excluded. Patients with adjuvant interferon-alpha treatments were included.

The tumors were fixed in 10% buffered formalin and embedded in paraffin (FFPE), and the samples were coded. Paraffin sections were routinely stained with hematoxylin and eosin, and a pathological diagnosis was made. Pathological stage and grade were determined according to the Union for international Cancer Control TNM classification of malignant tumors. Pathological grades were assigned according to the 2016 World Health Organization classification [47].

### 4.2. Immunohistochemistry

Antibodies against anti-ERK5 (abcam Japan, Tokyo, Japan) were used for immunohistochemistry (IHC). The staining was performed as previously described [48]. Briefly, a 5 μm-thick FFPE sample was mounted on silanized glass slides (Dako Japan, Tokyo, Japan). After deparaffination and rehydration, epitopes were reactivated by autoclaving the sections in a 10 mM citric buffer (pH 6.0) for 10 min. The slides were incubated with the primary antibodies and held overnight at 4 °C in a moist chamber. After washing with phosphate-buffered saline, the bound antibody was detected by the peroxidase method using Histofine simple stain MAZ-PO (Nichirei, Tokyo, Japan). The staining reaction was developed by diaminobenzidine in the presence of H_2_O_2_. Nuclear counterstaining was performed using hematoxylin. Positive and negative controls were included in each staining series.

Two investigators (SN and TN), who were both blinded to the patient data, evaluated the expression of EKR5 on tumor cells and scored them as strong, weak, and negative expression semi-quantitatively (Figure 1A). Any discrepancy was resolved by consensus.

### 4.3. TCGA Database

To validate the value as a prognostic indicator, we collected clinical data and the ERK5 mRNA expression level of Kidney Renal Clear Cell Carcinoma in the Cancer Genome Atlas (TCGA; https://cancergenome.nih.gov/, accessed on 28 June 2018) via cBioPortal (http://www.cbioportal.org/, accessed on 28 June 2018).

### 4.4. Cells and Culture Conditions

The established renal cell cancer cell lines A498, 786O, 769P, Caki1, and A704; the established human normal renal cortical epithelial cell line HRCEpC; and the established human endothelial cell line HUVEC were purchased from ATCC (Manassas, VA, USA). The cells were cultured as described previously [49].

### 4.5. Cell Viabitliy

Cell viability was detected with the CellTiter 96^®^ Aqueous One Solution Cell Proliferation Assay (Promega, Madison, WI, USA), as described previously [49]. ERK5 inhibitor, XMD8-92 and XMD17-109 were purchased from ChemoScene (Monmouth Junction, NJ, USA). Values of the half maximal inhibitory concentration (IC_50_) were calculated by fitting concentration–response curves to a four-parameter logistic sigmoidal function model using R package ‘drc’ (http://www.bioassay.dk; Copenhagen, Denmark).

### 4.6. Precursor microRNA Introduction

A498 cells were transfected with 30 nmol/L of precursor microRNA, pre-miR-143 (Applied Biosystems Japan, Tokyo, Japan). Transfection was performed with Lipofectamin 2000 (Thermo Fisher Scientific, Tokyo, Japan) as described previously [49].

### 4.7. MicroRNA Extraction and Real-Time PCR for Analysis of miR-143 Expression

Before microRNA extraction, fresh surgical specimens of paired malignant and normal renal tissue were immersed in an RNAlater (Applied Biosystems) tissue storage solution and stored at −80 °C until further use.

MicroRNA extraction and real-time PCR were performed with the mirVana^®^ miRNA Isolation Kit (Applied Biosystems), a TaqMan^®^ Universal PCR Master Mix (Applied Biosystems), and TaqMan^®^ MicroRNA Assays (has-miR-143 and RNU6B, Applied Biosystems), as previously described [49]. The expression of miRNAs was calculated using the comparative Ct (2-delta-delta Ct) method [50] with RNU6B as an endogenous control to normalize miRNAs expression levels. Each reaction was run in triplicate and means with SD were calculated. When the ratio of miR143 expression in malignant to normal tissues was under 0.8, the miR-143 expression was considered decreased.

### 4.8. RNA Interference

The transient knockdown of ERK5 was achieved in A498 cells using two siRNA (its sequence—TTTGCCTTACTTCCCACCTGtt and CCCATGTCGAAAGACTGGtt; Integrated DNA Technologies, Coralville, IA, USA). An unrelated control siRNA was also used (Applied Biosystems). Transfection was carried out using Lipofectamine max (Invitrogen, Waltham, MA, USA) according to the manufacturer’s recommendations.

### 4.9. Western Blotting Analysis

Western blotting analysis was performed as described previously [49]. The following antibodies were used: anti-βactin (Abcam, Cambridge, MA, USA), BCL2, p21, PARP, and ERK5 (Cell Signaling Technology Japan, Osaka, Japan).

### 4.10. Luciferase Reporter Assay

A luciferase reporter assay was performed as described previously [10]. The DNA plasmids were transfected into A498 cells using the transfection reagent Lipofectamine 3000 (Thermo Fisher Scientific). Briefly, the cells were seeded onto 12-well plates at 1 × 10^5^ (cells/well) and cultivated for a day. The DNA plasmids (3.3 µg of MEF2C/Luc, 0.1 µg of pGL4.74) and transfection reagents (5.0 µL/tube) were mixed gently in DMEM (100 µL/tube) and incubated for 15 minutes at room temperature. After the incubation, this entire mixture was transferred to the cultured media (100 µL/well). After incubation for 10 h at 37 °C, media were replaced with growth medium (500 µL) and XMD8-92 at the indicated concentration (for free, 1.25 µM, 2.5 µM, 5.0 µM, and 10 µM of XMD8-92). For reporter gene assays, the cells were incubated with XMD8-92 at 37 °C for 8 h. Cells were lysed in lysis buffer (Duel-Luciferase Reporter Asay System, Promega, Tokyo, Japan) (100 µL/well). After lysis, the luciferase reaction was induced by adding luciferase assay substrate, and luciferase activity was measured using a luminometer (Lumat LB 9507, Berthold Japan K.K., Tokyo, Japan). As an internal control, renillla luciferase activity was measured in the lysates to normalize for transfection efficiencies.

### 4.11. Apoptosis Assay

The apoptosis assay was performed as follows; A498 cells were cultured in the presence and absence of XMD8-92 for 72 h. The apoptosis was estimated by using 10 μM CellEvent^TM^ Caspase-3/7 Green Detection Reagent, which is a fluorogenic substrate for activated caspase-3/7 and 100 nM Invitrogen^TM^ tetramethylrhodamine ethyl ester (TMRE) (both products; Thermo Fisher Scientific, Tokyo, Japan), and this reflects the mitochondrial membrane’s potential.

### 4.12. Xenograft Tumor Model of ccRCC in Mice

All experimental procedures using female BALB/c-nu mice (CLEA Japan Inc., Tokyo, Japan) were performed according to the animal welfare regulations of Yamagata University Faculty of Medicine. Three to five mice per cage were maintained on a 12-hour light/dark cycle and were provided with sterilized water and standard rodent feed. Fourteen mice were randomized into two groups (7 mice in each group). A498 cells (8.5 × 10^5^) were resuspended in DMEM and injected subcutaneously into the right flank of 6-week-old female mice, as described previously [36]. When the tumor volume reached approximately 1000 mm^3^, the mice were treated with XMD8-92 (*n* = 6, 50 mg/kg of body weight; fifth a week) or vehicle 20% cyclodextrin (*n* = 5, control group), which was administrated by I.P. injections five times weekly for three weeks. One mouse in the XMD-8-92 group and two mice in another control group were excluded from the analysis because their tumor did not grow enough. Animal health was monitored daily and in the case of the appearance of a palpable tumor, its size was measured every 2–3 days with a caliper. The volume of tumors was calculated as (L × W^2^) × 0.5. We did not use the strategy to minimize potential confounders such as the order of treatment and measurements or animal cage location. The primary endpoint was the tumor volume after a 3-week treatment, which was set correspondingly to the design of this study. All mice were sacrificed by cervical dislocations under anesthesia using isoflurane after tumor volume assessments. Although we planned to euthanize mice with more than 20% body weight loss, we did not actually do so. After sacrifice, removed tumors were fixed in FFPE. The FFPE xenograft tumors were stained against Ki67 and CD34 (DAKO Japan, Tokyo, Japan) following the IHC procedure. In addition, one mouse was treated with 50 mg/kg of body XMD8-92 twice a day, five days a week for three weeks in order to examine the dose-dependency on the results of IHC.

### 4.13. Statistical Analysis

The statistical correlations between parameters of clinical background and ERK5 expression level were analyzed using Fisher’s exact test for categorical data and Welch’s *t* test for continuous variabilities. Survival durations were estimated using the Kaplan–Meier method and compared among groups using the log-rank test. The multivariate analysis was performed using the Cox proportional model. The tumor volume in mouse xenograft model was compared using one-way ANOVA. A *p*-value of <0.05 was considered statistically significant. All statistical analyses were performed using the statistical software package R version 3.5.2 (https://cran.r-project.org. accessed on 28 June 2018).

## 5. Conclusions

Our work shows that ERK5 is regulated by not only VHL inactivation, but also by miR-143 suppression in ccRCC. Our results also identify ERK5 as a potential therapeutic target in ccRCC.

## Figures and Tables

**Figure 1 ijms-23-08448-f001:**
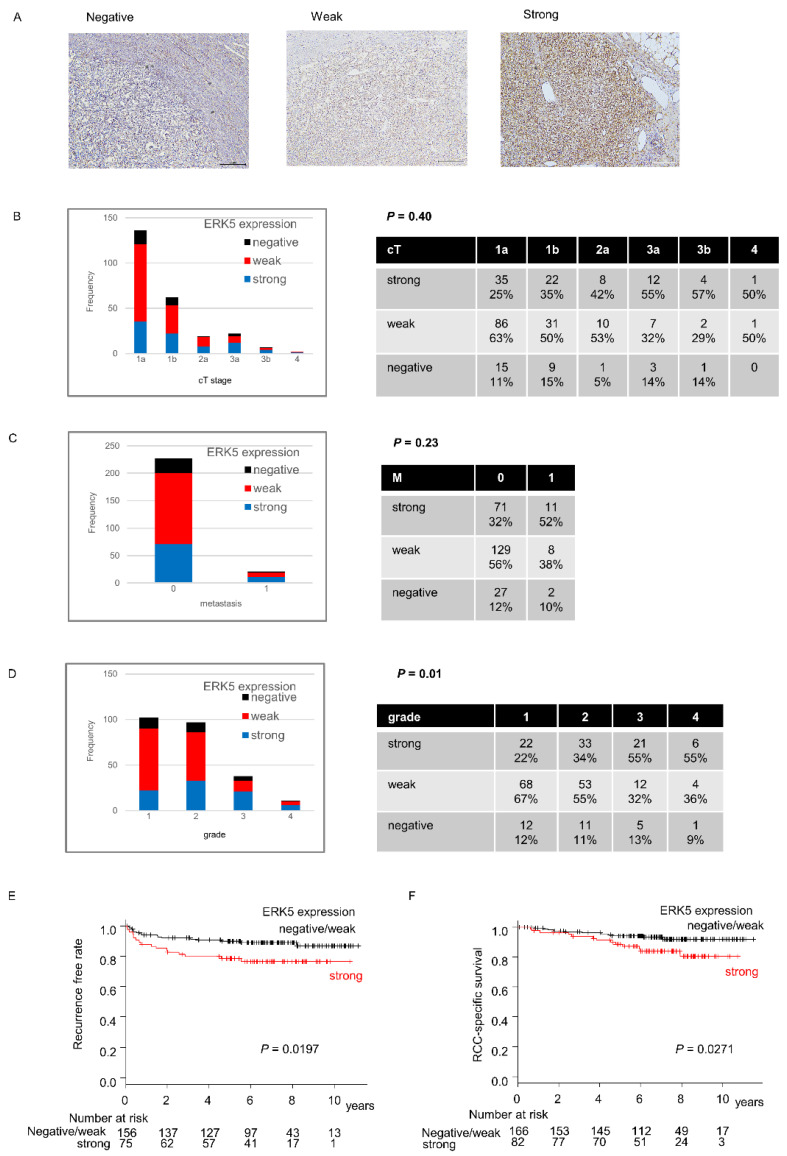
ERK5 expression in ccRCC specimens. (**A**) Representatives of negative, weak, and strong expression of ERK5 in immunohistochemistry specimens. (**B**) ERK5 expression level by each cT stage. (**C**) ERK5 expression level by each M stage. (**D**) ERK5 expression level by each grade. (**E**) Recurrence-free survival divided by ERK5 expression. (**F**) Cancer-specific survival divided by ERK5 expression. Abbreviations; ERK5, extracellular signal-regulated protein kinase 5; ccRCC, clear cell renal cell carcinoma.

**Figure 2 ijms-23-08448-f002:**
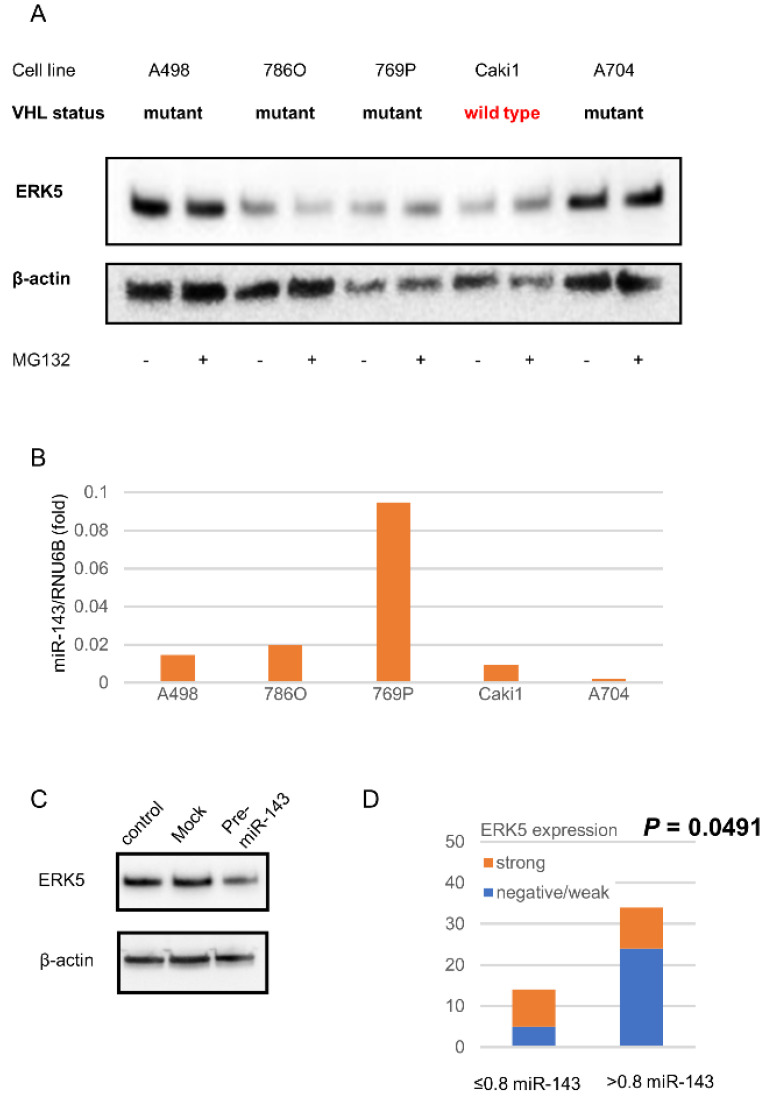
Correlations between ERK5 and miR-143. (**A**) ERK5 expression with/without proteasome inhibitor MG132 in RCC cells. The RCC cells were treated with 20 μM MG132 for 24 h. (**B**) miR-143 expression level in RCC cells. (**C**) Transduction of pre-miR-143 suppressed ERK5 expression. (**D**) miR-143 expression in surgical specimens. Abbreviation; ERK5, extracellular signal-regulated protein kinase 5; RCC, clear cell renal cell carcinoma.

**Figure 3 ijms-23-08448-f003:**
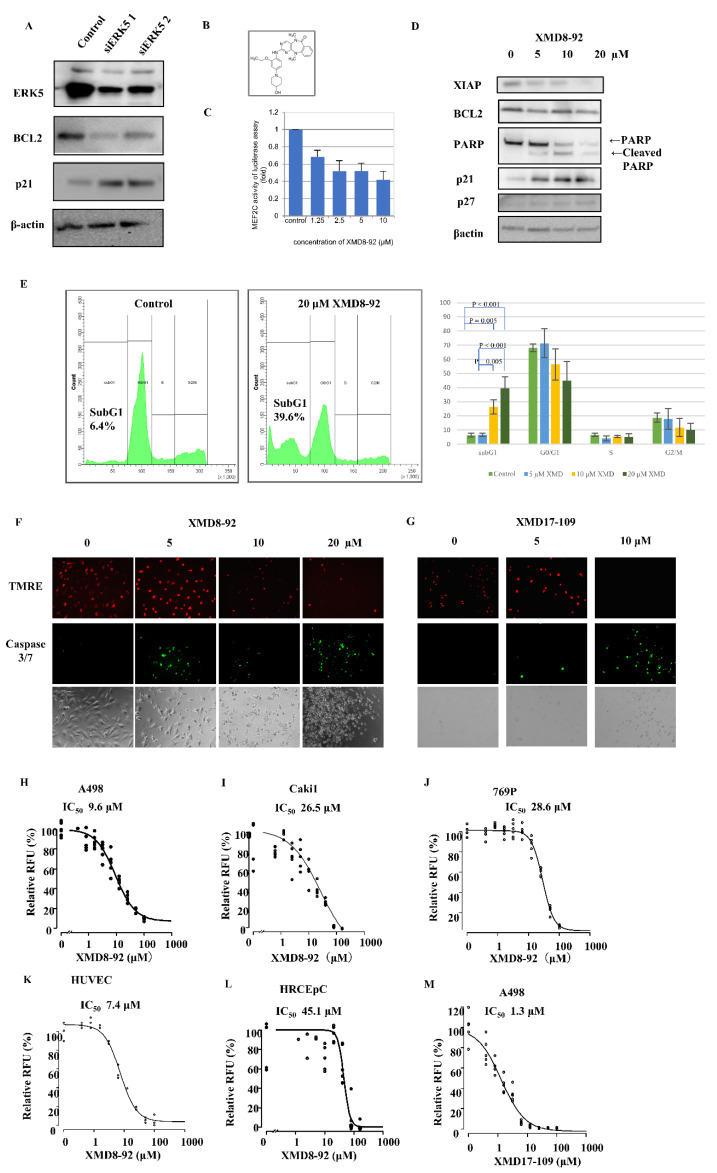
Effects of ERK5 inhibition. (**A**) Transient knockdown of ERK5 inhibited BCL2 and enhanced p21 on Western blotting in A498 cells. (**B**) The structure of ERK5 inhibitor XMD8-92. The error bars indicate standard deviation. (**C**) XMD8-92 suppressed the expression of MEF2C, ERK5, down-stream protein in a dose-dependent manner on luciferase-reporter assays in A498 cells. (**D**) XMD8-92 suppressed anti-apoptotic protein BCL2, produced cleaved PARP, and enhanced p21 and p27 on Western blotting in A498 cells. (**E**) Cell cycle analyses using flow cytometry with/without XMD8-92 in A498 cells. Flow cytometry was performed three times. The middle and left charts are the representative flow cytometry with/without XMD8-92. The right chart shows the average ratio of the cell-cycle distribution. The error bars indicate standard deviation. XMD8-92 augmented subG1 population in a dose-dependent manner. (**F**) The apoptosis assay with/without XMD8-92 for 24 h. (**G**) The apoptosis assay with/without XMD17-109 for 24 h. (**H**–**L**) MTS assay and IC_50_ with XMD8-92 in A498 (*VHL* mutant and low miR143 expression ccRCC cell line, (**H**)), Caki1 (*VHL* wild type ccRCC cell line, (**I**)), 769P (VHL mutant and high miR144 expression ccRCC cell line, (**J**)), HUVEC (human vascular endothelial cell line, (**K**)), and HRCEpC cells (normal renal cortical epithelial cell line, (**L**)). (**M**) MTS assay and IC_50_ with XMD17-109 in A498. Abbreviations: IC_50_, half maximal inhibitory concentration; ccRCC, clear cell renal cell carcinoma.

**Figure 4 ijms-23-08448-f004:**
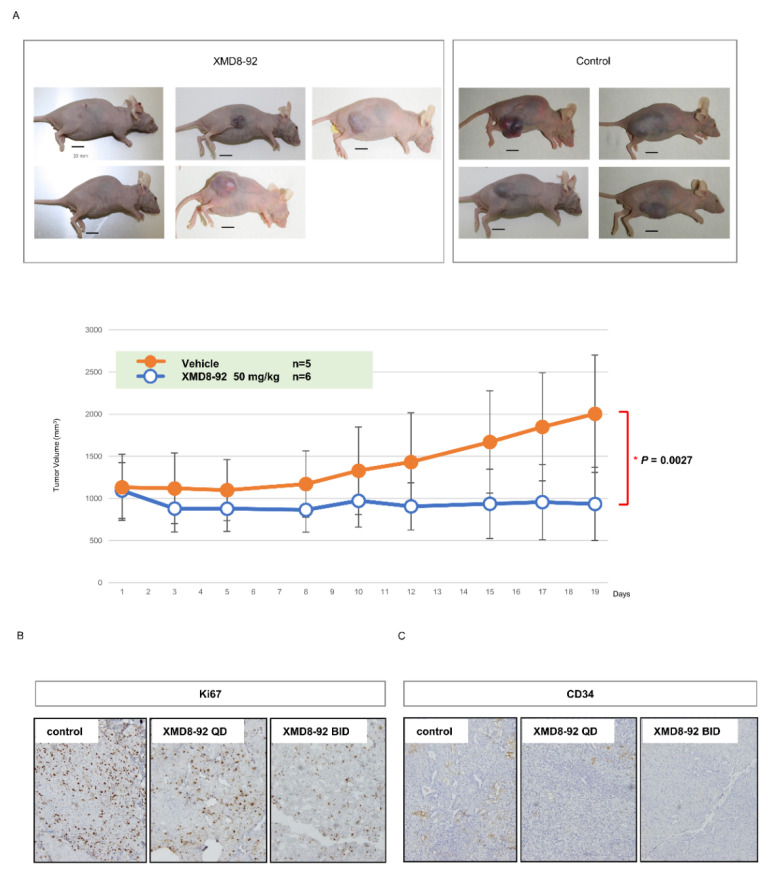
Subcutaneous A498 xenograft model with XMD8-92. (**A**) The mice were treated with XMD8-92 (*n* = 6, 50 mg/kg of body weight; fifth a week) or vehicle 20% cyclodextrin (*n* = 5) was administrated by I.P. injections five times weekly for 3 weeks. Data shown are mean ± SEM. The *p* value was determined by one-way ANOVA. (**B**,**C**) Mice were treated with XMD8-92 at a dose of 50 mg/kg once daily (QD), twice daily (BID), or with carrier solution intraperitoneally. Ki67 (B) and CD34 (C) expressions in the tumor were assessed by immunostaining.

**Table 1 ijms-23-08448-t001:** Patient Characteristics.

	Number	%
Age Median (Range)	64.7 (30.8–85.9)	
Sex		
Male	174	70.2
Female	74	29.8
cT		
1a	136	54.8
1b	62	25
2a	19	7.7
3a	22	8.9
3b	7	2.8
4	2	0.8
cN		
0	234	94.4
1	7	2.8
2	7	2.8
M		
0	227	91.5
1	21	8.5
Grade		
1	102	41.1
2	97	39.1
3	38	15.3
4	11	4.4
Outcome		
alive without RCC	181	73
alive with RCC	22	8.9
dead by RCC	24	9.7
dead by other cause	21	8.5

## Data Availability

The datasets used and/or analyzed during the current study are available from the corresponding author upon reasonable request.

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
