# Peer review of "Effect of Extracellular Signal-Regulated Protein Kinase 5 Inhibition in Clear Cell Renal Cell Carcinoma"

_ijms, 2022, doi:10.3390/ijms23158448_

Round 1

Reviewer 1 Report

Kano et al., demonstrated that ERK5 is highly expressed in patients with clear cell renal carcinoma, so its inhibition may be an interesting therapeutic approach. In the present work, a specific ERK5 inhibitor, XMD8-92 was tested in CCRC cell lines in vitro and in vivo, which induced apoptosis and increased of p21 and BCL-2 protein abundance. Mechanistically, the authors showed that ERK5 is controlled by miR143 in CCRC. Unfortunately, the study has limited novelty. As the regulation of ERK5 expression by miR143 has been reported before in breast and prostate cancer (PMID: 28746466; 23321517). 

I have several concerns that need to be address in order to be consider for publication.

1. MG-132 rescue experiment needs to be repeat, as the authors did not observed any effect from the proteasome inhibitor. The authors cited Arias-Gonzalez, et al 2013, and claimed their results were in accordance. However, the manuscript from 2013 showed that 20uM MG-132 for 5h induced the accumulation of ERK5 expression because impair the its degradation as a consequence of the ubiquitination. Authors should describe which concentration and time point were used to evaluate this effect. 

2. As the authors demonstrated that XM8-92 induced anti-proliferative effect of endothelial cells, I am wondering if any toxic effects were induced during drug treatment in vivo. 

2. How many times was the cell cycle experiment perform? Please provide standard deviations in the bar graph.

3. TMRE and caspase 3/7 images are too dark, so it is not possible to interpret the result.  Clear images need to be provide along with the quantification.

4. Label in the mice images are missing, are they showing tumor growth in different time-points?

5. Authors should evaluate the expression of other proteins involved in apoptosis, such as BAX, NOXA, PUMA., as p21 is a marker of cell cycle arrest and not of apoptosis. 

Author Response

Reviewer 1:

We appreciate your excellent suggestions. Our responses to your comments are listed  below.

  1. MG-132 rescue experiment needs to be repeat, as the authors did not observed any effect from the proteasome inhibitor. The authors cited Arias-Gonzalez, et al 2013, and claimed their results were in accordance. However, the manuscript from 2013 showed that 20uM MG-132 for 5h induced the accumulation of ERK5 expression because impair the its degradation as a consequence of the ubiquitination. Authors should describe which concentration and time point were used to evaluate this effect. 

Response: We thank the reviewer for indicating our miss that we did not write the MG132 treatment methods. We treated RCC cells with 20 μM MG132 for 24 hours. In Revised Manuscript, we added the treatment method into the Figure legend.

 Arias-Gonzalez, et al showed that MG132 induced the accumulation of ERK5 in Cos7, which is normal kidney cell line with wild-type VHL. They also showed pVHL introduced ERK5 degradation via ubiquitination/proteasome system. In addition, they mentioned 769P (VHL mutant ccRCC cell line) possess high ERK5 expression, comparing to Caki2 (VHL wild-type non-clear cell RCC cell line). In our article, we showed MG132 did not enhance ERK5 expression in VHL mutant ccRCC cells, while MG132 enhanced ERK5 expression in VHL wild type non clear cell RCC cell Caki1. We apologize for our insufficient explanation in our previous version. In Revised Manuscript, we added sentences into the “2.2. ERK5 and miR143 in ccRCC cell lines” in the result section as below; A previous report demonstrated that ERK5 accumulation is caused by impairment of pVHL/proteasome system [11]. To validate ERK5 accumulation in ccRCC cell lines, most of which harbor VHL inactivation, we examined ERK5 expression in VHL mutant ccRCC cell lines and a VHL wild type RCC cell line.

  1. As the authors demonstrated that XM8-92 induced anti-proliferative effect of endothelial cells, I am wondering if any toxic effects were induced during drug treatment in vivo. 

Response: Anti-vascular endothelial growth factor receptor (VEGFR) agents (i.e. sorafenib, sunitinib, pazopanib, axitinib, cabozantinib, and lenvatinib) are prevalently used in the field of ccRCC treatment. Anti-VEGFR agents should inhibit HUVEC cell proliferation. For example, we previously demonstrated that sorafenib inhibited HUVEC cell proliferation (Naito S, et al. Br J Cancer 2017, LDL cholesterol counteracts the antitumour effect of tyrosine kinase inhibitors against renal cell carcinoma; Figure 1D). Although VEGFR blockade could rationally impair renal and cardiac function due to endothelial damage, there are not many such adverse events in clinical practice (Escudier B, et al. J Clin Oncol 2009, Sorafenib for Treatment of Renal Cell Carcinoma: Final Efficacy and Safety Results of the Phase III Treatment Approaches in Renal Cancer Global Evaluation Trial).   

  1. How many times was the cell cycle experiment perform? Please provide standard deviations in the bar graph.

Response: Thank you for your kind comment. We did the cell cycle experiment three times. We added the SD bars in Figure 3E and the sentences into Figure legend as follows; The flow cytometry was performed three times. The middle and left charts are the representative flow cytometry with/without XMD8-92. The right chart shows the average ratio of the cell cycle distribution. The error bars indicate standard deviation.

  1. TMRE and caspase 3/7 images are too dark, so it is not possible to interpret the result.  Clear images need to be provide along with the quantification.

Response: Thank you for your kind comment. We changed to clear images in Figure 3F.

Reviewer 2 Report

In this manuscript, the authors tried to study the possible role of ERK5 in ccRCC. Although the scientific significance of ERK5 in ccRCC is high, the study cannot fully explaind the role of ERK5.

1. Although Fig 2C showed ERK5 can be suppressed by miR-143, the inverse correlation between ERK5 and miR-143 in ccRCC cell lines cannot be clearly observed from Fig 2A and 2B.  For example, miR-143 is very high in 769P, but ERK5 are comparable in 786O and 769P. In addition, the physiological effect of miR-143 on ERK5 still remains unclear.

2. In Fig 3 H - M, the authors demonstrated that XMD8-92 can inhibit several cell lines at different concentration. However, the authors failed to explain the reason for choosing the candidate cell lines.

Author Response

Reviewer 2:

We appreciate your excellent suggestions. Our responses to your comments are listed  below.

  1. Although Fig 2C showed ERK5 can be suppressed by miR-143, the inverse correlation between ERK5 and miR-143 in ccRCC cell lines cannot be clearly observed from Fig 2A and 2B.  For example, miR-143 is very high in 769P, but ERK5 are comparable in 786O and 769P. In addition, the physiological effect of miR-143 on ERK5 still remains unclear.

Response: We thank the reviewer for pointing out the important problem. The previous and our reports have demonstrated that pVHL and miR143 regulate the level of ERK5, however the low ERK5 level in 786O cannot be explained. There should be another mechanism regulating ERK5 expression, however, we have no idea on it. We added the sentences in discussion section as below; Meanwhile, there should be unknown mechanisms that regulate ERK5 other than pVHL and miR-143. Because 786O has low ERK5 expression, despite VHL mutant and low miR-143 expression level. Further studies will be needed to investigate other mechanisms of ERK5 regulation than pVHL and miR-143.

  1. In Fig 3 H - M, the authors demonstrated that XMD8-92 can inhibit several cell lines at different concentration. However, the authors failed to explain the reason for choosing the candidate cell lines.

Response: Thank you for your kind comment. We added the sentences into “2.4. ERK5 inhibition in ccRCC cells” as below; The high ERK5 expression ccRCC cell line A498 and endothelial cell HUVEC had lower IC50 than low ERK5 expression RCC cell line Caki1 and 769P and normal kidney cell line HRCEpC.

Round 2

Reviewer 1 Report

The authors addressed all the comments improving the quality of the manuscript, therefore it can be accepted for publication. 

Reviewer 2 Report

The authors have answered all concerns. English editing, however, is highly recommended.